# VECTORIZED CONDITIONAL NEURAL FIELDS: A FRAMEWORK FOR SOLVING TIME-DEPENDENT PDEs

**Jan Hagnberger**[1]**, Marimuthu Kalimuthu**[1,2,3]**, Mathias Niepert**[1,2,3]
Machine Learning and Simulation Lab, Institute for Artificial Intelligence (KI)
[1]University of Stuttgart, [2]Stuttgart Center for Simulation Science (SimTech)
[3]International Max Planck Research School for Intelligent Systems (IMPRS-IS)
`j.hagnberger@gmail.com`, {`firstname.lastname`}`@ki.uni-stuttgart.de`

## ABSTRACT

Transformer models are increasingly used for solving Partial Differential Equations (PDEs). However, they lack at least one of several desirable properties of an ideal surrogate model such as (i) generalization to PDE parameters not seen during training, (ii) spatial and temporal zero-shot super-resolution, (iii) continuous temporal extrapolation, (iv) applicability to PDEs of different dimensionalities, and (v) efficient inference for longer temporal rollouts. To address these limitations, we propose *Vectorized Conditional Neural Fields* (VCNeFs) which represent the solution of time-dependent PDEs as neural fields. Contrary to prior methods, VC-NeFs compute, for a set of multiple spatio-temporal query points, their solutions in parallel while also modeling their dependencies through attention mechanisms. Moreover, VCNeF can condition the neural field on both the initial conditions and the parameters of the PDEs. An extensive set of experiments demonstrates that VCNeFs are competitive with and often outperform existing ML-based surrogate models.

## 1 INTRODUCTION

The simulation of physical systems often involves solving PDEs and Machine Learning (ML) based surrogate models are increasingly used to address this challenging task (Lu et al., 2019; Li et al., 2020b; Cao, 2021). Utilizing ML methods for solving PDEs has several advantages such as faster simulation time than classical numerical PDE solvers, differentiability of the surrogate models (Takamoto et al., 2022), and their ability to be used even when the underlying PDEs are not known exactly (Li et al., 2020a).

Transformers (Vaswani et al., 2017) and its variants are successfully used in natural language processing (Devlin et al., 2018), speech processing (Gulati et al., 2020), and computer vision (Dosovitskiy et al., 2020). Additionally, the use of Transformer models in Scientific Machine Learning (SciML) to model physical systems (Geneva & Zabaras, 2020) and solve PDEs (Cao, 2021; Li et al., 2023a;b; McCabe et al., 2023) is steadily increasing. Meanwhile, recent advances in neural networks for computer graphics tasks have introduced Neural Fields (Xie et al., 2021), which have proven to be an efficient method to solve PDEs (Sitzmann et al., 2020; Chen et al., 2023b;a; Yin et al., 2023; Serrano et al., 2023).

Despite these recent advances in neural architectures for PDE solving, current methods lack several of the characteristics of an ideal PDE solver: (i) generalization to different Initial Conditions (ICs), (ii) PDE parameters, (iii) applicability to PDEs of different dimensionalities, (iv) stability over long rollouts, (v) temporal extrapolation, (vi) spatial and temporal super-resolution, all with affordable cost, high speed, and accuracy.

Towards developing a model that encompasses these ideal characteristics, we propose *Vectorized Conditional Neural Field* (VCNeF), a transformer-based conditional neural field, that solves PDEs continuously in time endowing the model with temporal as well as spatial Zero-Shot Super-Resolution (ZSSR) capabilities. The model introduces a new mechanism to condition a neural field

---

Code available at `https://github.com/jhagnberger/vcnef`

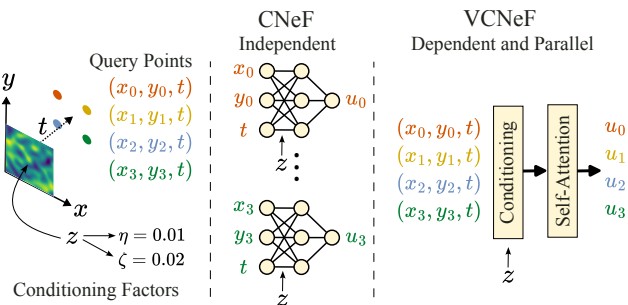

Figure 1: Conditional Neural Field (CNeF) vs proposed Vectorized CNeF (VCNeF).

on ICs and PDE parameters to achieve generalization to both ICs and PDE parameter values not seen during training. While modeling the solution using neural fields such as Physics-Informed Neural Networks (PINNS) naturally provides temporal and spatial ZSSR, these methods are inefficient since we need to query them separately for every spatial and temporal location. We achieve faster training and inference by vectorizing the computations on GPUs. Moreover, the proposed method also explicitly models dependencies between model queries for different spatial and temporal coordinates.

## 2 METHOD

In this section, we briefly recall (conditional) neural fields and relate them to solving parametric PDEs. Furthermore, we propose *Vectorized Conditional Neural Fields*.

**Neural Fields and Conditional Neural Fields.** Neural Fields (NeFs; Xie et al. (2021)) learn a function $f$ which maps the spatial and temporal coordinates (i.e., $\boldsymbol{x} \in \mathbb{R}^D, t \in \mathbb{R}_+$ respectively) to a quantity $\boldsymbol{q} \in \mathbb{R}^c$. More formally, a neural field can be expressed as a function

$$f_\theta : (\mathbb{R}_+ \times \mathbb{R}^D) \to \mathbb{R}^c \text{ with } (t, \boldsymbol{x}) \mapsto \boldsymbol{q} \tag{1}$$

that is parametrized by a neural network with parameters $\theta$. For solving PDEs, the function $f_\theta$ models the solution function $u$ and the quantity $\boldsymbol{q}$ represents the solution's value for the different channels, each representing a physical quantity (e.g., density). PINNs (Raissi et al., 2017) are a special case of neural fields with a physics-aware loss function, modeling the solution $u$ as

$$f_\theta : (\mathbb{R}_+ \times \mathbb{R}^D) \to \mathbb{R}^c \text{ with } (t, \boldsymbol{x}) \mapsto u(t, \boldsymbol{x}) \tag{2}$$

where $f_\theta$ denotes the neural field that maps the input spatial and temporal locations to the solution of the PDE. Conditional Neural Fields (CNeFs; Xie et al. (2021)) extend NeFs with a conditioning factor $\boldsymbol{z}$ to influence the output of the neural field. The conditioning factor was originally introduced for computer vision to control the colours or shapes of objects. In contrast, we condition the neural field, which models the solution of the PDE, on the IC and the PDE parameters (cf. Figure 1). Thus, the conditioning factor influences the entire field variable.

*Vectorized* **Conditional Neural Fields.** Typically, a (conditional) neural field generates the output quantities for all input spatial and temporal coordinates in multiple and independent forward passes. The training and inference times can be improved by processing multiple inputs in parallel on the GPU which is possible since all forward passes are independent and, hence, embarrassingly parallel. However, there are spatial dependencies between different input spatial coordinates, particularly for solving PDEs, that will not be exploited with CNeFs or by processing multiple inputs of CNeFs in parallel. Consequently, we propose extending CNeFs to take a vector with *arbitrary* spatial coordinates of *variable size* as input, exploit the dependencies among the input coordinates when generating the outputs, and regress all outputs for the inputs in one forward pass. Hence, we name our proposed model *Vectorized Conditional Neural Field* since it implicitly generates a vectorization of the input spatial coordinates for a given time $t$. In formal terms, VCNeF represents a function

$$f_\theta : (\mathbb{R}_+ \times \mathbb{R}^{s \times D}) \to \mathbb{R}^{s \times c} \text{ with } (t, X) \mapsto u(t, X) = \begin{pmatrix} u(t, \boldsymbol{x_1}) \\ \vdots \\ u(t, \boldsymbol{x_s}) \end{pmatrix} \tag{3}$$

| PDE | Model | nRMSE ($\downarrow$) | bRMSE ($\downarrow$) |
|-----|-------|---------|---------|
| Burgers | FNO | 0.0987 | 0.0225 |
| | MP-PDE | 0.3046 (+208.7%) | 0.0725 (+221.7%) |
| | CORAL | 0.2221 (+125.1%) | 0.0515 (+128.2%) |
| | Galerkin | 0.1651 (+67.3%) | 0.0366 (+62.3%) |
| | OFormer | 0.1035 (+4.9%) | **0.0215** (-4.5%) |
| | VCNeF | **0.0824** (-16.5%) | 0.0228 (+1.3%) |
| Advection | FNO | 0.0190 | 0.0239 |
| | MP-PDE | 0.0195 (+2.7%) | 0.0283 (+18.4%) |
| | CORAL | 0.0198 (+4.3%) | 0.0127 (-46.8%) |
| | Galerkin | 0.0621 (+227.1%) | 0.0349 (+46.2%) |
| | OFormer | **0.0118** (-38.0%) | **0.0073** (-69.6%) |
| | VCNeF | 0.0165 (-13.0%) | 0.0088 (-63.2%) |
| 1D CNS | FNO | 0.5722 | 1.9797 |
| | CORAL | 0.5993 (+4.7%) | 1.5908 (-19.6%) |
| | Galerkin | 0.7019 (+22.7%) | 3.0143 (+52.3%) |
| | OFormer | 0.4415 (-22.9%) | 2.0478 (+3.4%) |
| | VCNeF | **0.2943** (-48.6%) | **1.3496** (-31.8%) |
| 2D CNS | FNO | 0.5625 | 0.2332 |
| | Galerkin | 0.6702 (+19.2%) | 0.8219 (+252.4%) |
| | VCNeF | **0.1994** (-64.6%) | **0.0904** (-61.2%) |
| 3D CNS | FNO | 0.8138 | 6.0407 |
| | VCNeF | **0.7086** (-12.9%) | **4.8922** (-19.0%) |

Table 1: Errors of surrogate models trained and tested on the same spatial and temporal resolutions with a fixed PDE parameter value. *nRMSE* and *bRMSE* denote the normalized and RMSE at the boundaries, respectively. Values in parentheses indicate the percentage deviation to the FNO as a strong baseline in terms of accuracy, memory consumption, and runtime. Underlined values indicate the second-best errors.

where $u(t, \boldsymbol{x_i})$ denotes the PDE solution for the spatial coordinates $\boldsymbol{x_i}$. Note that we do not impose a structure on the spatial coordinates $\boldsymbol{x_i}$ and that the number of spatial points (i.e., $s$) can be arbitrary. The model can process multiple timesteps $t$ in parallel on the GPU to further improve the training and inference times since VCNeF does not exploit the dependencies between the temporal coordinates.

**VCNeF for Solving PDEs.** VCNeF allows direct learning of the solution function $u$ of a PDE instead of emulating a numerical PDE solver by mapping a timestep $t_n$ to a subsequent timestep $t_{n+1}$. The model is conditioned on the IC to allow for generalization to different ICs and on the PDE parameters $\boldsymbol{p}$ to generalize to PDE parameter values not seen during training. The VCNeF model can be expressed as a function

$$f_\theta : (\mathbb{R}_+ \times \mathbb{R}^{s \times D} \times \mathbb{R}^{s \times c} \times \mathbb{R}^j) \to \mathbb{R}^{s \times c} \text{ with } (t, X, u(0, X), \boldsymbol{p}) \mapsto u(t, X; u(0, X), \boldsymbol{p}), \quad (4)$$

where $\theta$ represents the parameters of the neural network, $X \in \mathbb{R}^{s \times D}$ the grid of dimensionality $D$ with query spatial coordinates, $t$ the query time, $u(0, X)$ the IC, and $\boldsymbol{p}$ the PDE parameter(s). $u(t, X; u(0, X), \boldsymbol{p})$ denotes the solution function, which depends on the given IC and PDE parameter(s), that is directly regressed by VCNeF. $\mathbb{R}^{s \times c}$ stands for the entire solution field with $c$ channels.

## 3 VCNeF FEATURES

**Spatial and Temporal ZSSR.** VCNeF can be trained on lower spatial and temporal resolutions and used for high-resolution spatial and temporal inference since it is, by virtue of its design, space and time continuous.

**Accelerated Training and Inference.** The training and inference of VCNeF are accelerated by processing multiple temporal coordinates in parallel on the GPU. If the solution of multiple timesteps (e.g., $t \in \{t_1, t_2, \ldots, t_{N_t}\}$) is to be predicted, VCNeF can calculate the solution of the timesteps

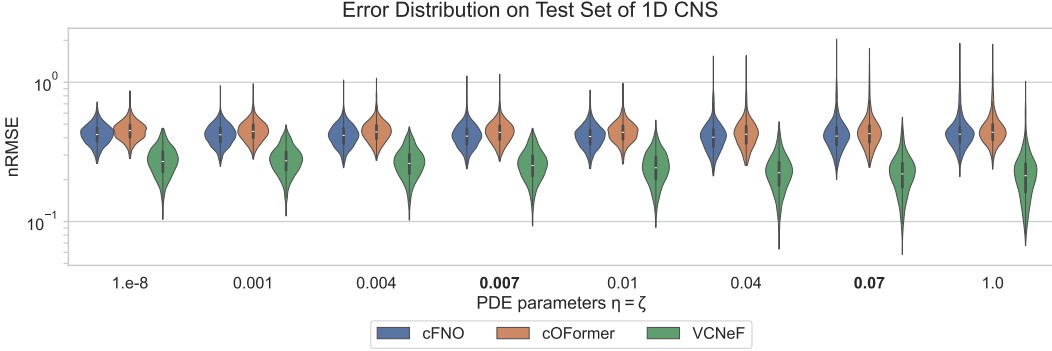

Figure 2: Error distribution of VCNeF vs. baselines. Boldfaced are unseen PDE parameter values.

in parallel since the predictions of $u(t, \cdot)$ are independent of each other and, hence, embarrassingly parallel.

**Physics-Informed VCNeF.** The loss function of VCNeF can be easily extended with a physics-informed loss as in PINNs (Raissi et al., 2017) since VCNeF directly models the solution function $u$ and, therefore, the derivatives can be computed with automatic differentiation (Maclaurin et al., 2015; Paszke et al., 2017).

## 4 EXPERIMENTS

We design our experiments such that we are able to answer the following research questions:

**RQ1**: How effective are VCNeFs compared to the SOTA methods for PDE solving? **RQ2**: How well can VCNeF generalize to PDE parameter values unseen during training? **RQ3**: How well can the model perform spatial and temporal zero-shot super-resolution? **RQ4**: Does the vectorization provide a speed-up, and what is the model's scaling behaviour when compared to the baselines?

### 4.1 DATASETS, SETUP, AND BASELINES

We conduct experiments on the following hydrodynamical equations of parametric PDEs from PDEBench (Takamoto et al., 2022): **1D Burgers'**, **1D Advection**, and **1D, 2D, and 3D Compressible Navier-Stokes** (CNS) equations. Unlike the prevalent models in the SciML literature, we train and test the models with a single timestep as IC and predicting multiple future steps, as this setting is best suited for real-world applications. We choose FNO (Li et al., 2020a), MP-PDE (Brandstetter et al., 2022), CORAL (Serrano et al., 2023), Galerkin Transformer (Cao, 2021), and OFormer (Li et al., 2023a) as the SOTA baselines. The predictions of FNO, Galerkin Transformer, and OFormer are achieved in an autoregressive fashion, while VCNeF predicts the entire trajectory of the simulation directly in one forward pass.

### 4.2 RESULTS

**RQ1.** We test VCNeF's generalization ability to different ICs by evaluating the models on the corresponding test sets of PDEBench. Table 1 shows the errors of the baseline and VCNeF models trained and tested on the selected PDEs and Appendix G contains example predictions. The results demonstrate that our model performs competitively with the SOTA methods for solving a variety of PDEs spatially ranging from 1D to 3D.

**RQ2.** To evaluate the performance and effectiveness of VCNeF and its PDE parameter conditioning, we train VCNeF, a PDE parameter conditioned FNO (cFNO; Takamoto et al. (2023)), and OFormer that has been modified to accept PDE parameter[1], on a set of PDE parameter values and

---

[1]We encode the parameter values as an additional channel.

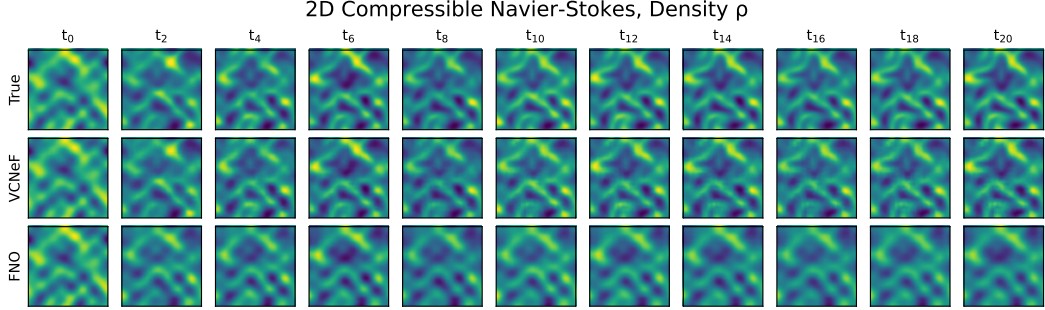

Figure 3: Example predictions of FNO and VCNeF for density channel of 2D CNS for $s = 64 \times 64$.

test them on an unseen set of parameter values. Figure 2 shows the error distribution for 1D CNS. We observe that VCNeF generalizes better to unseen PDE parameters than the compared baseline models.

**RQ3.** We evaluate the ability of VCNeF achieving spatial and temporal ZSSR which means that the model is trained on a reduced spatial and temporal resolution and then used for high-resolution inference. The results in Table 16 show that OFormer and our proposed VCNeF have spatial ZSSR capabilities since there is no significant increase in error even when the spatial resolution is 4x the training resolution. For the case of FNO on 1D Burgers, we observe a near multiplicative increase of error as the factor of resolution increases (2x and 4x). The inherent support of continuous-time inference of VCNeF allows predicting a trajectory with a smaller temporal step size than encountered during training. Table 17 shows a negligible increase in error for VCNeF exemplifying the model's superior temporal ZSSR capabilities by learning the dependency between the solution and time. Doing interpolation between timesteps seems to be only effective for smooth targets such as CNS.

**RQ4.** Lastly, we measure the inference times of the proposed and baseline models. The times are the raw inference times, without measuring the time needed to transfer the data from the host device to the GPU, on a single *NVIDIA A100-SXM4 80GB* GPU. We measure the time of the models predicting 40 to 240 timesteps in the future with a spatial resolution of $s = 256$. The times in Figure 7 demonstrate that VCNeF is significantly faster than the other transformer-based counterparts. However, the speed-up results in a higher GPU memory consumption as shown in Table 18.

## 5 CONCLUSION

In this work, we have designed an effective Neural PDE Solver, *Vectorized Conditional Neural Field*, based on the conditional neural fields framework, and demonstrated its generalization capabilities across multiple axes of desiderata: spatial, temporal, ICs, and PDE parameter values. Furthermore, the proposed model allows to trade-off the inference time or GPU memory when generating the solutions. As a future work, we aim to experiment on turbulent flow simulations and incompressible Navier-Stokes equations, improve the VCNeF model design further with adaptive timestepping, investigate sophisticated strategies for conditioning the neural fields (Rebain et al., 2023), test physics-informed losses (Li et al., 2021b), and analyze the model scaling behaviour on vast amounts of PDE data, including multiphysics simulations (McCabe et al., 2023; Sun et al., 2024).

### ACKNOWLEDGMENTS

Funded by Deutsche Forschungsgemeinschaft (DFG, German Research Foundation) under Germany's Excellence Strategy - EXC 2075 – 390740016. We acknowledge the support of Stuttgart Center for Simulation Science (SimTech). The authors thank IMPRS-IS for supporting Marimuthu Kalimuthu and Mathias Niepert. We extend our gratitude to Makoto Takamoto for the insightful discussions and Daniel Musekamp for his help with proofing the manuscript.

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

# Vectorized Conditional Neural Fields: A Framework for Solving Time-dependent Partial Differential Equations
## Supplementary Material

## A    RELATED WORK

Here we outline the work related to our study by categorizing them into four sections as follows.

### A.1    PHYSICS-INFORMED NEURAL NETWORKS AND NEURAL OPERATORS

A common approach for solving PDEs are Physics-Informed Neural Networks (Raissi et al., 2017) that model the underlying solution function $u$. Although PINNs are space-and-time continuous within the specified domain, they are finite-dimensional and hence cannot perform temporal extrapolation. Additionally, PINNs generate the solution for all input spatial coordinates independently without further exploitation of structural dependencies. A PDE-specific loss function allows the model to learn the underlying solution function which satisfies the PDE equation. However, PINNs can still fail to approximate the PDE solutions because of complex loss landscapes (Krishnapriyan et al., 2021). In contrast, neural operators (Li et al., 2020b) learn a mapping between two infinite-dimensional spaces or two functions, representing the solution function for a timestep $t_n$ and the subsequent timestep $t_{n+1}$. Consequently, neural operators are not continuous in time, but continuous in space and generate the solution for all spatial coordinates in a single forward pass (Lu et al., 2019). They further leverage the spatial dependencies of the solution by processing the spatial coordinates in one forward pass. The Fourier Neural Operator (FNO; Li et al. (2020a)) is a prevalent instantiation of a neural operator that is based on Fourier transforms. However, the FNO model is limited to regular meshes due to the use of the discrete Fourier transform, which is not suitable for data that was collected on irregular geometries or a sphere (e.g., climate data). To tackle this issue, Bonev et al. (2023) propose Spherical Fourier Neural Operators (SFNO) that generalize FNO (Li et al., 2020a) to deal with data on the sphere. Recent work of Guibas et al. (2022) improves FNO by combining the FNO and Transformer model (Vaswani et al., 2017) yielding Adaptive Fourier Neural Operator (AFNO) that replaces the self-attention mechanism of Transformers with a token mixing mechanism in the Fourier domain. The AFNO outperforms vanilla Transformers on several tasks and has a quasi-linear time complexity and linear memory and, thus, is more efficient compared to vanilla Transformers. Physics-Informed Neural Operators (PINO) (Li et al., 2023c) extend neural operators with a PDE-specific loss term to further improve the accuracy of the model. The proposed VCNeF can be seen as a combination of PINNs and neural operators.

### A.2    TRANSFORMERS FOR SOLVING PDES

Transformers are increasingly being utilized for modelling physical systems or PDEs. Previous works can be divided into using Transformers for applying temporal self-attention (Geneva & Zabaras, 2020) to model the temporal dependencies or applying spatial self-attention for capturing spatial dependencies of the PDE (Cao, 2021; Li et al., 2023a;b). Applying the spatial self-attention as in Fourier and Galerkin Transformer (Cao, 2021), or in OFormer (Li et al., 2023a) yields a neural operator (Kovachki et al., 2021) endowing the model with spatial ZSSR capabilities. Since these models do not consider time as an additional input, they are not time-continuous and, hence, fixed to a trained temporal discretization. To remedy the issue, the diffusion-inspired temporal Transformer operator (Ovadia et al., 2023) uses the time component to condition the input solution, thereby supporting a flexible temporal discretization for inference.

### A.3    SOLVING PARAMETRIC PDES

Although ML-based methods have shown great success in solving PDEs, they often do not consider PDE parameters as input resulting in failures to generalize to unseen parameter values. Recent works such as CAPE (Takamoto et al., 2023), MP-PDE (Brandstetter et al., 2022), and PDERefiner (Lippe et al., 2023) consider the PDE parameters as additional model input. Along the same lines, our proposed model also considers the PDE parameter to improve the generalization error of unseen PDE parameter values.

### A.4   Implicit Neural Representations (INR)

Neural Fields (NeFs) has become widely popular in signal processing (Sitzmann et al., 2020), computer vision (Mescheder et al., 2019), computer graphics (Chu et al., 2022), and recently in SciML for solving PDEs (Chen et al., 2023b; Yin et al., 2023; Chen et al., 2023a; Serrano et al., 2023).

The prevalent INR models for PDE solving follow the "Encode-Process-Decode" paradigm. DiNO (Yin et al., 2023) has an encoder, a Neural ODE (NODE) to model dynamics, and a decoder. CORAL, an improvement to DiNO, has a two-step training procedure whereby the input and output INR modules with shared parameters are trained first, and subsequently, the dynamics modeling block is trained using the learned latent codes. DiNO and CORAL utilize an INR to encode and decode the PDE solution in a latent space and a NODE to propagate the dynamics in latent space, while our approach utilizes a neural field to represent the entire PDE solution encompassing both the spatial and temporal dependencies within a shared space.

### A.5   Comparison of Neural Architectures for PDE Solving

Table 2 shows the most important properties of ML models for solving PDEs and compares three families of models. The VCNeF combines both worlds of neural fields (e.g., PINNs) and neural operators. PINNs do not leverage the spatial dependencies among the queried coordinates since they produce the output for each queried coordinate independently. Meanwhile, neural operators map to a set of solution points and leverage the dependencies between the regressed points. However, neural operators are usually not time-continuous, while PINNs are time-continuous. VCNeF, therefore, combines the advantages of both worlds by leveraging spatial dependencies by generating a set of solution points and being continuous in time.

| Model family | Model | Initial value generalization | PDE parameter generalization | ZSSR Spatial | ZSSR Temporal | Models spatial dependencies with self-attention |
|---|---|---|---|---|---|---|
| Neural Operator | FNO | ✓ | ✗ | ✓ | ✗ | ✗ |
|  | OFormer | ✓ | ✗ | ✓ | ✗ | ✓ |
|  | cFNO | ✓ | ✓ | ✓ | ✗ | ✗ |
|  | cOFormer | ✓ | ✓ | ✓ | ✗ | ✓ |
| Neural Field | PINN | ✗ | ✗ | ✓ | ✓ | ✗ |
| Conditional Neural Field | CORAL | ✓ | ✗ | ✓ | ✓ | ✗ |
|  | VCNeF (ours) | ✓ | ✓ | ✓ | ✓ | ✓ |

Table 2: Overview of distinct properties of benchmark baselines and our proposed VCNeF model.

## B   Neural Architecture

We propose a transformer-based VCNeF that applies self-attention to the spatial domain to capture dependencies between the spatial coordinates. The input spatio-temporal coordinates and the physical representation of ICs are represented in a latent space. Both latent representations are fed into modulation blocks that capture spatial dependencies and condition the coordinates on the IC. The output of the modulation blocks, which represent the solution, is then decoded to obtain the representation in the physical space.

### B.1   Model Components

**Latent Representation of Coordinates.**   The input coordinates, consisting of the query time $t \in \mathbb{R}_+$ that determines the time for which the model's prediction is sought and the spatial coordinates $\boldsymbol{x_i} \in \mathbb{R}^D$, are represented in a latent space. For 1D PDEs, a linear layer is used for encoding, whereas, for 2D and 3D PDEs, the absolute positional encoding (PosEnc; Vaswani et al. (2017)) to encode time $t$, similar to Ovadia et al. (2023) and learnable Fourier features (LFF; Li et al. (2021a))

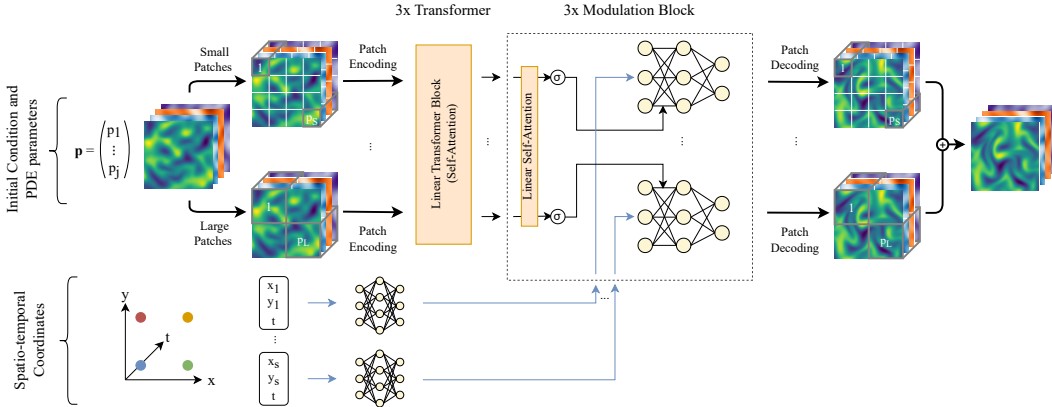

Figure 4: VCNeF for solving parametric PDEs. Latent representations of ICs are generated with a multi-scale patching mechanism (Chen et al., 2021). A Modulation Block consists of self-attention, activation function $\sigma$, and a modulated neural field which uses the pointwise multiplication (scaling) of FiLM (Perez et al., 2018) to condition the spatio-temporal coordinates on ICs.

to encode the spatial coordinates are used.

$$
\begin{aligned}
&\text{For 1D: } c_i = (t \parallel \boldsymbol{x_i})W + b \\
&\text{For 2D: } c_i = (\texttt{PosEnc}(t) \parallel \texttt{LFF}(\boldsymbol{x_i}) \parallel \ldots \parallel \texttt{LFF}(\boldsymbol{x_{i+15}}))W + b \\
&\text{For 3D: } c_i = (\texttt{PosEnc}(t) \parallel \texttt{LFF}(\boldsymbol{x_i}) \parallel \ldots \parallel \texttt{LFF}(\boldsymbol{x_{i+63}}))W + b \\
&\quad\texttt{LFF}(\boldsymbol{x}) = \frac{1}{\sqrt{d}}(\cos(\boldsymbol{x}W_r) \parallel \sin(\boldsymbol{x}W_r))^T \\
&\quad\quad \boldsymbol{C} = (c_1 \parallel \ldots \parallel c_s)^T
\end{aligned}
\tag{5}
$$

where $\parallel$ denotes the concatenation of two vectors.

**Latent Representation of IC.** The input IC is mapped to a latent representation by either applying a shared linear layer to each solution point $u(t, x_i)$ or by dividing the spatial domain into non-overlapping patches and applying a linear layer to the patches, akin to Vision Transformers (ViTs; Dosovitskiy et al. (2020)). However, unlike a traditional ViT, our patch generation has two branches: patches of a smaller size ($4 \times 4$) and of a larger size ($16 \times 16$) as proposed in Chen et al. (2021) since we aim to capture the dynamics accurately with multi-scale processing.

$$
\begin{aligned}
&\text{For 1D: } z_i^{(0)} = (u(t, \boldsymbol{x_i}) \parallel \boldsymbol{x_i} \parallel \boldsymbol{p})W + b \\
&\text{For 2D: } z_i^{(0)} = (u(t, \boldsymbol{x_i}) \parallel \ldots \parallel u(t, \boldsymbol{x_{i+15}}) \parallel \\
&\quad\quad\quad \boldsymbol{x_i} \parallel \ldots \parallel \boldsymbol{x_{i+15}} \parallel \boldsymbol{p})W + b \\
&\text{For 3D: } z_i^{(0)} = (u(t, \boldsymbol{x_i}) \parallel \ldots \parallel u(t, \boldsymbol{x_{i+63}}) \parallel \\
&\quad\quad\quad \boldsymbol{x_i} \parallel \ldots \parallel \boldsymbol{x_{i+63}} \parallel \boldsymbol{p})W + b \\
&\quad \boldsymbol{Z^{(0)}} = (z_1^{(0)} \parallel \ldots \parallel z_s^{(0)})^T
\end{aligned}
\tag{6}
$$

A vector in the latent space (i.e., token) either represents the solution on a spatial point (for 1D) or the solution on a patch of spatial points (for 2D and 3D). The grid contains the coordinates where the solutions are sampled in the spatial domain. This information is used when generating the latent representations of the IC to ensure that each latent representation has information about the position. The PDE parameters $\boldsymbol{p}$ are also added to the latent representation. We neglect additional positional encoding to prevent any length generalization problems (Ruoss et al., 2023) that could prevent changing the spatial resolution after training.

**Linear Transformer Encoder for IC.** We utilize a Linear Transformer (Katharopoulos et al., 2020) with self-attention in our VCNeF architecture to generate an attention-refined latent representation of the IC $\boldsymbol{Z^{(0)}}$. The global receptive field of Transformer allows the proposed architecture

to capture spatial dependencies in the IC, although each token contains only partial spatial information. Intuitively, the Transformer outputs latent representations that incorporate the entire spatial solution and not only a single spatial point or a subset of spatial points. We assume that this is beneficial to generate a better representation of the IC to condition the input spatio-temporal coordinates accordingly.

$$Z^{(n+1)} = \texttt{Transformer\_Block}\left(Z^{(n)}\right) \tag{7}$$

where $\texttt{Transformer\_Block}(\cdot)$ is a Linear Transformer block with self-attention and $n$ denotes the $n^{th}$ block.

**Modulation of Coordinates based on IC.** The modulation blocks condition the input coordinates on the input IC $Z^{(3)}$ by modulating the latent representation $C$ of the coordinates. The block contains self-attention, a non-linearity $\sigma$, a modulation mechanism similar to Feature-wise Linear Modulation (FiLM; Perez et al. (2018)), layer normalization, residual connections, and an MLP. However, the conditioning mechanism uses only the scaling (i.e. pointwise multiplication) of FiLM and omits the shift (i.e. pointwise addition). A modulation block is expressed succinctly as

$$\begin{aligned}
Z^{(m+1)} &= \texttt{Modulation\_Block}\left(C, Z^{(m)}\right) \\
&= \texttt{MLP}\left(\sigma\left(\texttt{Self\_Attn}\left(Z^{(m)}\right)\right) \circ \texttt{MLP}(C)\right) \\
\sigma(X) &= \texttt{ELU}(X) + 1
\end{aligned} \tag{8}$$

where $Z^{(3)} \in \mathbb{R}^{s \times d}$ represents the IC, $C \in \mathbb{R}^{s \times d}$ denotes the latent representation of the input coordinates, $\circ$ represents the Hadamard product, and $m$ is the $m^{th}$ fusion block. The residual connections and layer normalization are omitted in Equation 8 for the sake of simplicity.

**Decoding the Solution's Latent Representation.** The solution's latent representation $Z^{(6)}$ is mapped back to the physical space by either applying an MLP for 1D or by mapping the latent representations to small and large patches and outputting the weighted sum of the small and large patches for 2D and 3D.

## B.2 Ablation Study

We conduct an ablation study on the prominent parts of the proposed architecture. Namely, the self-attention mechanism that allows the model to capture spatial dependencies and the conditioning mechanism that is used to condition the neural field. For 2D PDEs, we also study the effect on the model's performance for patches of one size and the multi-scale patching mechanism with small and large patches. Additionally, we compare linear attention (Katharopoulos et al., 2020) with vanilla attention (Vaswani et al., 2017) in terms of training time and GPU memory consumption. For simplicity, we perform the ablation study mainly on the 1D Burgers' and 2D CNS datasets.

**Self-Attention, Conditioning Mechanism, and Patch Generation.** Table 3 shows the results of the proposed model with and without self-attention as well as with different modulation mechanisms to condition the neural field. Table 4 presents the different results for patches of one size vs multi-scale patching mechanism.

| PDE | Self-attention | Conditioning mechanism | nRMSE ($\downarrow$) | bRMSE ($\downarrow$) |
|:---:|:---:|:---:|:---:|:---:|
| | ✓ | Modulation with scaling | **0.0824** | **0.0228** |
| 1D Burgers | ✗ | Modulation with scaling | 0.8890 | 0.3242 |
| | ✓ | Modulation with scaling and shifting | 0.0945 | 0.0291 |

Table 3: Ablation study for *attention* and *conditioning* mechanisms of our proposed VCNeF model.

**Vanilla Attention and Linear Attention.** The Linear Transformer and the linear self-attention component in the proposed architecture can be replaced with vanilla attention or some arbitrary attention mechanism. We choose linear attention since it promises a speed-up for long sequences (i.e.,

| PDE | Patches | nRMSE ($\downarrow$) | bRMSE ($\downarrow$) |
|---|---|---|---|
| 2D CNS | Small and large | **0.1994** | **0.0904** |
| | Only large | 0.45694 | 0.19822 |

Table 4: Ablation study for the multi-scale patching mechanism of our proposed VCNeF model.

fine resolution of the spatial domain) compared to vanilla attention. Table 5 shows empirical results for training the transformer-based VCNeF on the 1D Burgers' PDE. We observe that the memory of linear attention increases linearly and of vanilla attention quadratically. Double the spatial resolution corresponds to double the number of tokens yielding an increased memory and time consumption. Training the VCNeF with vanilla attention requires more than 640 GiB while the VCNeF with linear attention requires only 99.4 GiB. We use the vanilla attention implementation of Katharopoulos et al. (2020) for a fair comparison to the non-optimized linear attention implementation.

| PDE | Spatial resolution (# tokens) | Attention type | GPU memory | Time per epoch |
|---|---|---|---|---|
| 1D Burgers | 256 | Vanilla | 72.6 GiB | 28 s |
| | | Linear | 31.4 GiB | 18 s |
| | 512 | Vanilla | 223.4 GiB | 78 s |
| | | Linear | 53.8 GiB | 32 s |
| | 1024 | Vanilla | >640 GiB | N/A |
| | | Linear | 99.4 GiB | 62 s |

Table 5: GPU memory consumption and training time per epoch of VCNeF with vanilla attention (scaled dot-product attention) and linear attention on the 1D Burgers' train set. The values refer to training with a batch size of 64 on 4x *NVIDIA A100-SXM4 80GB* GPUs using data parallelism. The number of queried timesteps $N_t$ is 40. *Time per epoch* includes the time that is needed to load the data from a network share and transfer it to the GPUs.

## C  PDE DATASETS

This part serves as an expository section on the parametric PDE datasets of the hydrodynamical equations that we experimented with to answer our research questions.

### C.1  1D BURGERS' EQUATION

The Burgers' PDE models the non-linear behaviour and diffusion process in fluid dynamics and is expressed as

$$\partial_t u(t, x) + u(t, x)\partial_x u(t, x) = \frac{\nu}{\pi}\partial_{xx} u(t, x) \qquad (9)$$

kinematic viscosity $\nu$

where the PDE parameter $\nu$ denotes the diffusion coefficient. Our dataset contains solutions for x $\in$ (-1, 1) with a maximum resolution of 1024 spatial discretization points and t $\in$ (0, 2] with a maximum resolution of 201 temporal discretization steps including the initial condition. We subsample the data along the temporal and spatial domain yielding a trajectory of 41 time steps where each snapshot has a spatial resolution of 256, respectively.

### C.2  1D ADVECTION EQUATION

The Advection PDE models pure advection behaviour without non-linearity. It is written as

$$\partial_t u(t, x) + \beta\, \partial_x u(t, x) = 0 \qquad (10)$$

advection velocity $\beta$

where the PDE parameter $\beta$ denotes the advection velocity. Similar to 1D Burgers', we subsample the data to get samples with a trajectory of 41 time steps each with a spatial resolution of 256.

### C.3   1D, 2D, AND 3D COMPRESSIBLE NAVIER-STOKES EQUATIONS

Following Takamoto et al. (2022; 2023), we conduct experiments on the challenging Compressible Navier-Stokes datasets, with random initial fields as the initial conditions for 1D, 2D, and 3D cases.

$$\partial_t \rho + \nabla \cdot ( \rho \, \mathbf{v}) = 0, \tag{11a}$$

(mass density (mass per unit volume) — label for $\rho$)

$$\rho(\partial_t \mathbf{v} + \mathbf{v} \cdot \nabla \mathbf{v}) = -\nabla p + \eta \triangle \mathbf{v} + ( \zeta + \frac{\eta}{3})\nabla(\nabla \cdot \mathbf{v}), \tag{11b}$$

(shear viscosity — label for $\eta$; bulk viscosity — label for $\zeta$)

$$\partial_t(\epsilon + \frac{\rho v^2}{2}) + \nabla \cdot [(p + \epsilon + \frac{\rho v^2}{2})\mathbf{v} - \mathbf{v} \cdot \sigma'] = \mathbf{0}, \tag{11c}$$

(viscous stress tensor — label for $\sigma'$; internal energy — label for $\epsilon$)

where $\rho$ represents the (mass) density of the fluid, $\mathbf{v}$ the fluid velocity (in vector field), p the gas pressure, $\epsilon$ the internal energy according to the equation of state, $\sigma'$ the viscous stress tensor, $\eta$ and $\zeta$ the PDE parameters which represent the shear and bulk viscosities, respectively. We subsample the data for 1D, 2D, and 3D equations. The original resolution of 1D simulation has 1024 spatial points and 101 timesteps for each trajectory. For training purposes, we subsample across both spatial and temporal resolutions by a factor of 4 and 2 respectively yielding a trajectory of length 51 time steps and a spatial resolution of 256. To be consistent with other 1D PDE trajectory lengths, we retain only the first 41 timesteps and perform experiments on this temporally truncated data. The original resolution of the 2D simulation data is $128 \times 128$ for each channel (i.e., density, velocity-x, velocity-y, and pressure) and has 21 timesteps. For training, we have subsampled the data only for the spatial dimensions resulting in a resolution of $64 \times 64$. For 3D data, the original spatial resolution is $128 \times 128 \times 128$ which was subsampled to a resolution of $32 \times 32 \times 32$ for training.

## D   BASELINE MODELS

### D.1   FOURIER NEURAL OPERATOR

Fourier Neural Operator (FNO; Li et al. (2020a)) is an implementation of a neural operator that maps from one function $a(x)$ to another function $a'(x)$ (Kovachki et al., 2021). Traditionally, neural networks learn a mapping between two finite-dimensional Euclidean spaces which leads to the problem that they are fixed to a spatial resolution when used for solving PDEs. Neural Operators overcome this limitation by learning an *operator* that is a mapping between infinite-dimensional function spaces (i.e., mapping between functions). The function $a(x) := u(t_n, \boldsymbol{x})$ represents the solution for timestep $t_n$ and $a'(x) := u(t_{n+1}, \boldsymbol{x})$ the solution for a future timestep $t_{n+1}$. FNO, an instantiation of a neural operator, is based on spectral convolution layers which implement an integral transformation of the input function. The integral transformation is implemented with discrete Fourier transforms on the spatial domain allowing an efficient and expressive architecture. We use the FNO model as a baseline because it is a very strong model in terms of accuracy, speed, and memory consumption. The FNO model is trained in an autoregressive fashion for 500 epochs.

### D.1.1 cFNO

cFNO (Takamoto et al., 2023) is the adapted version of the FNO where the PDE parameters are added as an additional channel to condition the model on the PDE parameters.

## D.2 MP-PDE: Message Passing PDE Solvers

Several models exist that use Graph Neural Networks (GNNs) for solving PDEs (Brandstetter et al., 2022; Boussif et al., 2022)

**Message Passing Neural PDE Solvers.** MP-PDE (Brandstetter et al., 2022) follows the prevalent *Encode-Process-Decode* framework for simulating physical systems (Sanchez-Gonzalez et al., 2020). The MP-PDE model has an MLP as encoder, GNN as a processor, and a CNN as the decoder. Moreover, the model introduces several tricks such as pushforward, temporal bundling (time window with 5 timesteps), and random timesteps in the length of the trajectory as starting points during training for autoregressive PDE solving, while also considering the PDE parameter values as additional input, making it as a versatile choice for generalized neural PDE solving. Hence, we adopt it as a baseline. However, it has to be noted that we apply the model only to 1D PDEs. The configuration of our adaptation for 1D PDEs amounts to 614,929 model parameters which is comparable to the other baselines.

## D.3 CORAL: Coordinate-based Model for Operator Learning

Considering that CORAL (Serrano et al., 2023), to the best of our knowledge, is the current state-of-the-art INR-based method for solving PDEs, we benchmark our proposed VCNeF model against it on 1D Advection and Burgers' as well as on the challenging 1D and 2D compressible Navier-Stokes PDEs (Takamoto et al., 2022). The CORAL model is trained purely in a data-driven manner and involves two stages: (i) INR training, and (ii) Dynamics Modelling training. Due to the sequential nature of this two-phase training, first, we train the INR model and dynamics model is trained after the completion of INR model training. CORAL authors conducted experiments on a small dataset of 256 training and 16 test samples. We, on the other hand, conduct experiments on PDEBench which consists of 9000 train and 1000 test samples. Hence, we train the CORAL baseline model for 1000 epochs for INR training and 500 epochs for dynamics modelling optimization, unlike the original authors' suggested setting of 10000 epochs of optimization for both INR and dynamics modelling training.

As in the case of other baseline models, we train and test the CORAL baseline on the subsampled data yielding a spatial and temporal resolution of 256 and 41 respectively. We report results on 1D Advection, Burgers', and CNS. The training of 2D CNS resulted in very high errors in the INR training phase and the loss diverged to NaN values in dynamics modeling training. Hence, we exclude 2D results. We encode both the single and multiple channel inputs of PDEs in a single latent space of dimension 256 with the aim to keep the model simple and match the number of parameters to other baseline models. For other hyperparameter values such as the learning rate, NODE depth and width, we use the default values suggested by Serrano et al. (2023).

## D.4 Galerkin Transformer

Cao (2021) introduces the novel application of self-attention for learning a neural operator. The author provides an alternative way to interpret the matrices $Q, K, V$ by interpreting them column-wise as the evaluation of learned basis functions instead of row-wise as the latent representation of the tokens. This new interpretation allows the author to improve the effectiveness of the attention mechanism by linearizing it, yielding Fourier and Galerkin-type Attention. The author employs the proposed attention mechanisms in a transformer-based neural operator for solving PDEs. We choose Galerkin Transformer as a baseline because it is transformer-based and uses self-attention on the spatial domain of the PDE. The baseline model is trained for 500 epochs in an autoregressive fashion using the hyper-parameters suggested by Cao (2021).

### D.5 OFORMER

OFormer (Li et al., 2023a) is a transformer-based neural operator which is based on the Galerkin and Fourier Transformer (Cao, 2021). Existing approaches such as FNO and Galerkin or Fourier Transformer are restricted in having the same grid for the input and output. Consequently, it is not possible to query the model (i.e., output function) on arbitrary spatial points that are different or partially disjoint from the input points. OFormer solves this problem by adding cross-attention to the model to allow querying for arbitrary spatial points. In addition, Li et al. (2023a) suggest further improvements to the Galkerin or Fourier Transformer and name the resulting model Operator Transformer (OFormer). We train the OFormer model in an autoregressive fashion with the curriculum learning strategy of Takamoto et al. (2023) for 500 epochs.

#### D.5.1 cOFORMER

Inspired by cFNO (Takamoto et al., 2023) we adapt OFormer to take the PDE parameter as an additional input. The PDE parameter values are appended to the input as an additional channel to condition the model on the PDE parameter.

## E ADDITIONAL EXPERIMENTAL DETAILS

### E.1 USED PDE PARAMETERS

Unlike the prevalent models in the SciML literature, we train and test the models with a single timestep as IC and predicting multiple future steps. We use the PDE parameter values in Table 6 for our experiments with one fixed PDE parameter value, i.e., we train and test the models on the same parameter.

| PDE | Timesteps | Spatial res. | PDE parameters |
|---|---|---|---|
| 1D Burgers | 41 | 256 | $\nu = 0.001$ |
| 1D Advection | 41 | 256 | $\beta = 0.1$ |
| 1D CNS | 41 | 256 | $\eta = \zeta = 0.007$ |
| 2D CNS | 21 | $64 \times 64$ | $\eta = \zeta = 0.01$ |
| 3D CNS | 11 | $32 \times 32 \times 32$ | $\eta = \zeta = 10^{-8}$ |

Table 6: Fixed PDE parameters used in our experiments with a fixed PDE parameter value.

Table 7 shows the combinations of PDE parameter values used in our experiments for the multiple parameters setting. In this case, we train the models on a set of PDE parameter values (**seen**) and test it on a different set of PDE parameter values (**unseen**) with the aim to test the model's generalization capabilities on this aspect.

| PDE | Training Set Parameters (seen) | Test Set Parameters (unseen) |
|---|---|---|
| 1D Burgers | $\nu = (0.002, 0.004, 0.02, 0.04, 0.2, 0.4, 2.0)$ | $\nu = (0.001, 0.01, 0.1, 1.0, 4.0)$ |
| 1D Advection | $\beta = (0.2, 0.4, 0.7, 2.0, 4.0)$ | $\beta = (0.1, 1.0, 7.0)$ |
| 1D CNS | $\eta = \zeta = (10^{-8}, 0.001, 0.004, 0.01, 0.04, 0.1)$ | $\eta = \zeta = (0.007, 0.07)$ |

Table 7: Exemplary set of PDE parameters used in our experiments with multiple PDE parameters.

### E.2 MODEL'S HYPERPARAMETERS

Tables 8, 9 and 10 list the used hyperparameters for the baselines and our proposed VCNeF model.

| PDE | Model | Epochs | Batch size | Fourier width | # Fourier Modes | # Layers | Learning rate | LR Scheduler | # Parameters | Curriculum learning |
|---|---|---|---|---|---|---|---|---|---|---|
| 1D Burgers | FNO | 500 | 64 | 64 | 16 | 4 | 1.e-4 | Step Scheduler | 549,569 | ✗ |
| 1D Advection | FNO | 500 | 64 | 64 | 16 | 4 | 1.e-4 | Step Scheduler | 549,569 | ✗ |
| 1D CNS | FNO | 500 | 64 | 64 | 16 | 4 | 6.e-5 | Step Scheduler | 549,955 | ✗ |
| 2D CNS | FNO | 500 | 64 | 32 | 12 | 4 | 3.e-4 | Step Scheduler | 9,453,716 | ✗ |
| 3D CNS | FNO | 500 | 4 | 20 | 12 | 4 | 3.e-4 | Step Scheduler | 22,123,753 | ✗ |

Table 8: Hyperparameters for the FNO used in the single PDE parameter experiments. The Step Scheduler was configured with a step size of 100 and gamma of 0.5.

| PDE | Model | Epochs | Batch size | Embedding size | # Layers | Time window | Learning rate | # Parameters | Curriculum learning |
|---|---|---|---|---|---|---|---|---|---|
| 1D Burgers | MP-PDE | 20 | 64 | 128 | 6 | 5 | 1.e-4 | 614,929 | ✗ |
| 1D Advection | MP-PDE | 20 | 64 | 128 | 6 | 5 | 1.e-4 | 614,929 | ✗ |

Table 9: Hyperparameters for the MP-PDE used in the single PDE parameter experiments.

| PDE | Model | Epochs | Batch size | Embedding size | # Heads | # Layers | Learning rate | LR Scheduler | # Parameters | Curriculum learning |
|---|---|---|---|---|---|---|---|---|---|---|
| 1D Burgers | OFormer | 500 | 64 | 96 | 1 | 4+3 | 6.e-5 | One Cycle Scheduler | 660,814 | ✓ |
|  | Galerkin | 500 | 64 | 96 | 1 | 4+2 | 1.e-5 | One Cycle Scheduler | 530,305 | ✗ |
|  | VCNeF | 500 | 32 | 96 | 8 | 3+3 | 3.e-4 | One Cycle Scheduler | 793,825 | ✗ |
| 1D Advection | OFormer | 500 | 64 | 96 | 1 | 4+3 | 6.e-5 | One Cycle Scheduler | 660,814 | ✓ |
|  | Galerkin | 500 | 64 | 96 | 1 | 4+2 | 1.e-5 | One Cycle Scheduler | 530,305 | ✗ |
|  | VCNeF | 500 | 64 | 96 | 8 | 3+3 | 6.e-4 | One Cycle Scheduler | 793,825 | ✗ |
| 1D CNS | OFormer | 500 | 64 | 96 | 1 | 4+3 | 6.e-5 | One Cycle Scheduler | 662,733 | ✓ |
|  | Galerkin | 500 | 64 | 96 | 1 | 4+2 | 1.e-5 | One Cycle Scheduler | 530,595 | ✗ |
|  | VCNeF | 500 | 64 | 96 | 8 | 3+3 | 4.e-4 | One Cycle Scheduler | 794,307 | ✗ |
| 2D CNS | VCNeF | 1000 | 64 | 256 | 8 | 1+6 | 3.e-4 | One Cycle Scheduler | 11,779,436 | ✗ |
| 3D CNS | VCNeF | 1000 | 4 | 256 | 8 | 1+6 | 3.e-4 | One Cycle Scheduler | 27,335,041 | ✗ |

Table 10: Hyperparameters for the transformer-based models used in the single PDE parameter experiments. The One Cycle Scheduler was configured to reach the maximum learning rate at 0.2, start division factor 1.e-3 and final division factor 1.e-4.

### E.3 EVALUATION METRICS

We use the normalized RMSE (nRSME) and boundary RMSE (bRMSE) from PDEBench (Takamoto et al., 2022) as metrics to evaluate the models.

**Normalized RMSE (nRMSE).** The normalized RMSE ensures the independence of the different scales. The channels of PDEs with multiple channels are often on different scales (e.g., one channel consists of values with small magnitudes while another channel consists of values with large magnitudes). Additionally, the scale of a single channel usually changes when the time-dependent PDE evolves in time (e.g., large magnitudes at the beginning of the trajectory decaying to small magnitudes at the end). nRMSE is independent of these scaling effects and provides a good metric for the global and local performance of the ML model. Let $y \in \mathbb{R}^{N_t \times s \times c}$ be the ground truth trajectory and $\hat{y} \in \mathbb{R}^s$ the model's prediction where $N_t$ denotes the length, $s = s_x \cdot s_y \cdot \ldots$ the spatial points per timestep, and $c$ the number of channels of the PDE. Then, the per-sample nRMSE is defined as

$$\text{relativeError}(t, c_i) = \frac{\|y_{t,\cdot,c_i} - \hat{y}_{t,\cdot,c_i}\|_2}{\|y_{t,\cdot,c_i}\|_2} \qquad \in \mathbb{R}$$

$$\text{nRMSE} = \frac{1}{N_t \cdot c} \sum_{t=1}^{N_t} \sum_{i=1}^{c} \text{relativeError}(t, i) \qquad \in \mathbb{R} \tag{12}$$

**Boundary RMSE (bRMSE).** The RMSE on the boundaries of the spatial domain quantifies whether the boundary condition can be learned or not. Let $y \in \mathbb{R}^{N_t \times s_x \times c}$ be the ground truth trajectory of a 1D PDE and $\hat{y} \in \mathbb{R}^{N_t \times s_x \times c}$ the model's prediction where $N_t$ denotes the length, $s_x$ denotes the number of points for the x-axis, and $c$ the number of channels of the PDE under consideration. Then, the per-sample bRMSE is defined as

$$\text{boundaryError}(t, c_i) = \sqrt{\frac{(y_{t,1,c_i} - \hat{y}_{t,1,c_i})^2 + (y_{t,s,c_i} - \hat{y}_{t,s,c_i})^2}{2}} \qquad \in \mathbb{R}$$

$$\text{bRMSE} = \frac{1}{N_t \cdot c} \sum_{t=1}^{N_t} \sum_{i=1}^{c} \text{boundaryError}(t, i) \qquad \in \mathbb{R} \tag{13}$$

## F ADDITIONAL EXPERIMENTAL RESULTS

This section contains additional results of the experiments. We train all models on two different initializations and provide the mean and standard deviations of the runs. Similar to the main section in the main paper, we structure the results to answer the four research questions.

### F.1 (RQ1): COMPARISON WITH BASELINES

The Tables 11, 12, 13, 14, and 15 show the metrics with standard deviations for the chosen PDEs and models.

| Model | nRMSE ($\downarrow$) | bRMSE ($\downarrow$) |
|---|---|---|
| FNO | $0.0987^{\pm 0.0004}$ | $0.0225^{\pm 0.0006}$ |
| MP-PDE | $0.3046^{\pm 0.0004}$ | $0.0725^{\pm 0.0014}$ |
| CORAL | $0.2221^{\pm 0.0108}$ | $0.0515^{\pm 0.0001}$ |
| Galerkin | $0.1651^{\pm 0.0044}$ | $0.0366^{\pm 0.0012}$ |
| OFormer | $0.1035^{\pm 0.0059}$ | $\mathbf{0.0215}^{\pm 0.0009}$ |
| VCNeF | $\mathbf{0.0824}^{\pm 0.0004}$ | $0.0228^{\pm 0.0003}$ |

Table 11: Normalized RMSE (nRMSE) and RMSE at the boundaries (bRMSE) of baselines and proposed model for the 1D Burgers' equation with $\nu = 0.001$.

| Model | nRMSE ($\downarrow$) | bRMSE ($\downarrow$) |
|---|---|---|
| FNO | $0.0190^{\pm 0.0003}$ | $0.0239^{\pm 0.0002}$ |
| MP-PDE | $0.0195^{\pm 0.0011}$ | $0.0283^{\pm 0.0022}$ |
| CORAL | $0.0198^{\pm 0.0031}$ | $0.0127^{\pm 0.0014}$ |
| Galerkin | $0.0621^{\pm 0.0024}$ | $0.0349^{\pm 0.0011}$ |
| OFormer | $\mathbf{0.0118}^{\pm 0.0012}$ | $\mathbf{0.0073}^{\pm 0.0008}$ |
| VCNeF | $0.0165^{\pm 0.0007}$ | $0.0088^{\pm 0.0003}$ |

Table 12: Normalized RMSE (nRMSE) and RMSE at the boundaries (bRMSE) of baselines and proposed model for the 1D Advection equation with $\beta = 0.1$.

| Model | nRMSE ($\downarrow$) | bRMSE ($\downarrow$) |
|---|---|---|
| FNO | $0.5722^{\pm 0.0244}$ | $1.9797^{\pm 0.0029}$ |
| CORAL | $0.5993^{\pm 0.1014}$ | $1.5908^{\pm 0.1341}$ |
| Galerkin | $0.7019^{\pm 0.0002}$ | $3.0143^{\pm 0.0112}$ |
| OFormer | $0.4415^{\pm 0.0115}$ | $2.0478^{\pm 0.0581}$ |
| VCNeF | $\mathbf{0.2943}^{\pm 0.0034}$ | $\mathbf{1.3496}^{\pm 0.0254}$ |

Table 13: Normalized RMSE (nRMSE) and RMSE at the boundaries (bRMSE) of baselines and proposed model for the 1D CNS equation with $\eta = \zeta = 0.007$.

| Model | nRMSE ($\downarrow$) | bRMSE ($\downarrow$) |
|---|---|---|
| FNO | $0.5625^{\pm 0.0015}$ | $0.2332^{\pm 0.0001}$ |
| Galerkin | $0.6702^{\pm 0.0036}$ | $0.8219^{\pm 0.0043}$ |
| VCNeF | $\mathbf{0.1994}^{\pm 0.0086}$ | $\mathbf{0.0904}^{\pm 0.0036}$ |

Table 14: Normalized RMSE (nRMSE) and RMSE at the boundaries (bRMSE) of baselines and proposed model for the 2D CNS equation with $\eta = \zeta = 0.01$.

| Model | nRMSE ($\downarrow$) | bRMSE ($\downarrow$) |
|---|---|---|
| FNO | $0.8138^{\pm 0.0007}$ | $6.0407^{\pm 0.0493}$ |
| VCNeF | $\mathbf{0.7086}^{\pm 0.0005}$ | $\mathbf{4.8922}^{\pm 0.0077}$ |

Table 15: Normalized RMSE (nRMSE) and RMSE at the boundaries (bRMSE) of baselines and proposed model for the 3D CNS equation with $\eta = \zeta = 10^{-8}$.

## F.2 (RQ2): GENERALIZATION TO UNSEEN PDE PARAMETER VALUES

We test VCNeF's generalization capabilities to unseen PDE parameter values by training it on a set of PDE parameter values and testing it on a different set of unseen PDE parameter values. We use cFNO (Takamoto et al., 2023) and cOFormer as the state-of-the-art baselines. Both models have been adapted to encode the PDE parameter as an additional input channel. Figures 5 and 6 show the error distribution over the corresponding PDE parameter values and test sets.

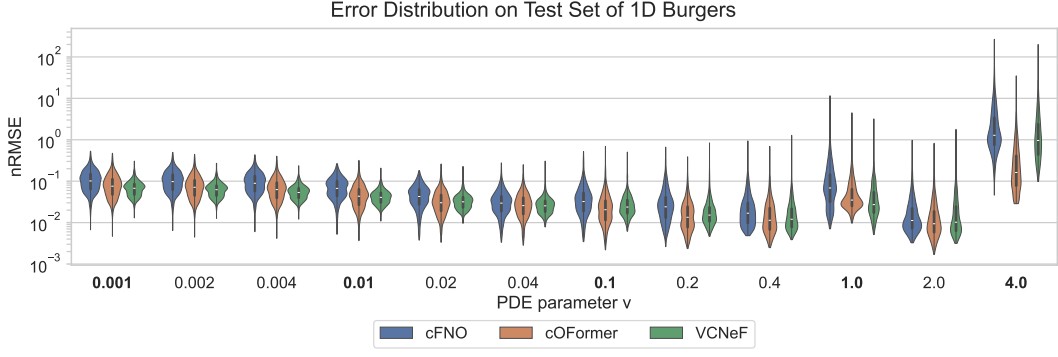

Figure 5: Error distribution of samples in the test set of 1D Burgers. Boldfaced are the unseen PDE parameter values.

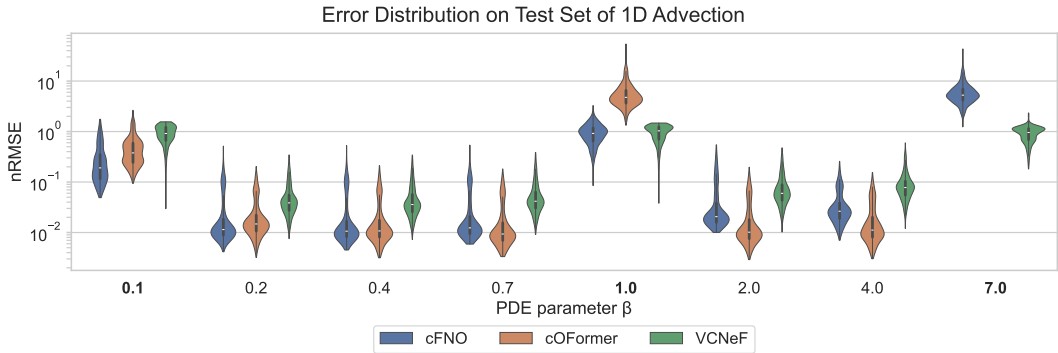

Figure 6: Error distribution of samples in the test set of 1D Advection. Boldfaced are the unseen PDE parameter values. Values for cOFormer and $\beta = 7.0$ are missing since the model produced NaN at inference time.

## F.3 (RQ3): Temporal and Spatial ZSSR

Here we discuss the spatial and temporal zero-shot super-resolution capabilities of VCNeF and compare it with FNO and OFormer for spatial ZSSR. As for the temporal ZSSR, we chose CORAL since it is a continuous-time model, and FNO+Interpolation because FNO does not support temporal ZSSR. Table 16 shows the errors for the spatial ZSSR experiment and Table 17 for the temporal ZSSR experiment.

| PDE | Spatial resolution $s$ | Model | nRMSE ($\downarrow$) | bRMSE ($\downarrow$) |
|---|---|---|---|---|
| Burgers | 256 | FNO | 0.0987 | 0.0225 |
| | | OFormer | 0.1035 | **0.0215** |
| | | VCNeF | **0.0824** | 0.0228 |
| | 512 | FNO | 0.2557 | 0.0566 |
| | | OFormer | 0.1092 | **0.0228** |
| | | VCNeF | **0.0832** | 0.0229 |
| | 1024 | FNO | 0.3488 | 0.0766 |
| | | OFormer | 0.1102 | 0.0233 |
| | | VCNeF | **0.0839** | **0.0230** |
| 1D CNS | 256 | FNO | 0.5722 | 1.9797 |
| | | OFormer | 0.4415 | 2.0478 |
| | | VCNeF | **0.2943** | **1.3496** |
| | 512 | FNO | 0.6610 | 2.7683 |
| | | OFormer | 0.4657 | 2.5618 |
| | | VCNeF | **0.2943** | **1.3502** |
| | 1024 | FNO | 0.7320 | 3.5258 |
| | | OFormer | 0.4655 | 2.5526 |
| | | VCNeF | **0.2943** | **1.3510** |
| 2D CNS | $64 \times 64$ | FNO | 0.5625 | 0.2332 |
| | | VCNeF | **0.1994** | **0.0904** |
| | $128 \times 128$ | FNO | 0.8693 | 2.3944 |
| | | VCNeF | **0.4016** | **0.2280** |
| 3D CNS | $32 \times 32 \times 32$ | FNO | 0.8138 | 6.0407 |
| | | VCNeF | **0.7086** | **4.8922** |
| | $64 \times 64 \times 64$ | FNO | 0.9452 | 8.7068 |
| | | VCNeF | **0.7228** | **5.1495** |
| | $128 \times 128 \times 128$ | FNO | 1.0077 | 9.8633 |
| | | VCNeF | **0.7270** | **5.3208** |

Table 16: Errors of spatial ZSSR experiment. The models are trained on the spatial resolutions indicated in grey and tested on higher spatial resolutions. The temporal resolution is $N_t = 41$ for 1D, $N_t = 21$ for 2D, and $N_t = 11$ for 3D (same as during training). Underlined values indicate the second-best errors.

| PDE | Temporal resolution $N_t$ | Model | nRMSE ($\downarrow$) | bRMSE ($\downarrow$) |
|---|---|---|---|---|
| Burgers | 41 | FNO | 0.0987 | **0.0225** |
| | | CORAL | 0.2221 | 0.0515 |
| | | VCNeF | **0.0824** | 0.0228 |
| | 101 | FNO + Interp. | 0.1116 | 0.0279 |
| | | CORAL | 0.5298 | 0.1682 |
| | | VCNeF | **0.0829** | **0.0234** |
| | 201 | FNO + Interp. | 0.1154 | 0.0294 |
| | | CORAL | 0.6186 | 0.2013 |
| | | VCNeF | **0.0831** | **0.0236** |
| Advection | 41 | FNO | 0.0190 | 0.0239 |
| | | CORAL | 0.0198 | 0.0127 |
| | | VCNeF | **0.0165** | **0.0088** |
| | 101 | FNO + Interp. | 0.0234 | 0.0242 |
| | | CORAL | 0.8970 | 0.4770 |
| | | VCNeF | **0.0165** | **0.0088** |
| | 201 | FNO + Interp. | 0.0258 | 0.0247 |
| | | CORAL | 0.9656 | 0.5376 |
| | | VCNeF | **0.0165** | **0.0088** |
| 1D CNS | 41 | FNO | 0.5722 | 1.9797 |
| | | CORAL | 0.5993 | 1.5908 |
| | | VCNeF | **0.2943** | **1.3496** |
| | 82 | FNO + Interp. | 0.5667 | 1.9639 |
| | | CORAL | 1.1524 | 3.7960 |
| | | VCNeF | **0.2965** | **1.3741** |
| 3D CNS | 11 | FNO | 0.8138 | 6.0407 |
| | | VCNeF | **0.7086** | **4.8922** |
| | 21 | FNO + Interp. | 0.8099 | 6.1938 |
| | | VCNeF | **0.7106** | **5.1446** |

Table 17: Errors for temporal ZSSR. The models are trained on the temporal resolutions indicated in grey and tested on higher temporal resolutions. The spatial resolution is $s = 256$ for 1D and $s = 64 \times 64 \times 64$ for 3D (same as during training). "FNO + Interp." means FNO with linear interpolation between the timesteps since it doesn't naturally support temporal ZSSR. Underlined values indicate the second-best errors.

### F.4 (RQ4): INFERENCE TIMES

In traditional numerical solvers, the simulation time of trajectories of a given PDE is influenced by several factors such as the value of PDE parameter, efficiency of software implementation, the type and order of numerical algorithm, fineness of discretization meshes, length of the domain, and so on. On the contrary, inference (simulation) time of ML models is agnostic to these factors, which is one of the huge advantages of machine-learned PDE surrogates.

To compare and understand the time consumed for longer rollouts, we visualize the inference times of VCNeF, FNO, Galkerin Transformer, and OFormer models. Figure 7 shows the scaling behaviour of the inference times for a variable number of timesteps in the future {80, 120, 160, 200, 240}. Table 18 includes the memory consumption in addition to the inference times. The results demonstrate that the inference times of the proposed VCNeF model scale better when compared to other transformer-based baselines and are competitive with FNO. However, the speed-up results in a higher memory consumption. The model can also be used to do inference in a sequential fashion which reduces the memory consumption but increases the inference time. Nevertheless, it is still faster than OFormer while the memory requirement remains the same even for extended rollout durations making it as a strong contender among the transformer-based neural PDE solvers.

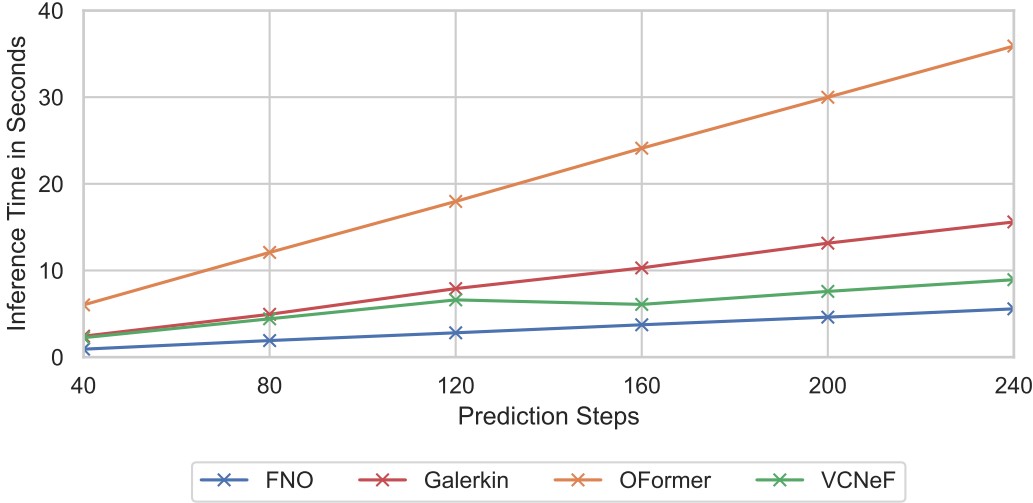

Figure 7: Inference times of the listed models predicting different numbers of timesteps in the future with a fixed spatial resolution of $s = 256$ on 1D Burgers.

| Prediction Steps | Model | Inference Time [ms] | | GPU Memory Consumption [MiB] |
|---|---|---|---|---|
| 40 | FNO | 917.77 | $\pm 2.51$ | 716 |
| | Galerkin | 2415.99 | $\pm 54.56$ | 632 |
| | OFormer | 6025.75 | $\pm 12.75$ | 990 |
| | VCNeF | 2244.04 | $\pm 6.65$ | 4724 |
| | VCNeF sequential | 4853.17 | $\pm 75.29$ | 644 |
| 80 | FNO | 1912.19 | $\pm 56.03$ | 716 |
| | Galerkin | 4940.80 | $\pm 89.44$ | 632 |
| | OFormer | 12081.98 | $\pm 19.39$ | 990 |
| | VCNeF | 4422.65 | $\pm 4.11$ | 9284 |
| | VCNeF sequential | 9701.80 | $\pm 84.48$ | 644 |
| 120 | FNO | 2808.04 | $\pm 82.22$ | 716 |
| | Galerkin | 7908.18 | $\pm 96.52$ | 644 |
| | OFormer | 17965.47 | $\pm 14.19$ | 988 |
| | VCNeF | 6606.41 | $\pm 3.00$ | 13638 |
| | VCNeF sequential | 14577.00 | $\pm 112.83$ | 644 |
| 160 | FNO | 3733.10 | $\pm 62.94$ | 716 |
| | Galerkin | 10295.78 | $\pm 116.50$ | 644 |
| | OFormer | 24108.24 | $\pm 6.45$ | 990 |
| | VCNeF | 6084.04 | $\pm 9.37$ | 18871 |
| | VCNeF sequential | 19449.80 | $\pm 113.73$ | 644 |
| 200 | FNO | 4614.21 | $\pm 97.52$ | 718 |
| | Galerkin | 13151.47 | $\pm 93.95$ | 644 |
| | OFormer | 29986.81 | $\pm 6.35$ | 990 |
| | VCNeF | 7584.48 | $\pm 1.86$ | 22328 |
| | VCNeF sequential | 24252.38 | $\pm 101.41$ | 644 |
| 240 | FNO | 5572.07 | $\pm 109.23$ | 716 |
| | Galerkin | 15600.60 | $\pm 262.51$ | 644 |
| | OFormer | 35900.51 | $\pm 6.71$ | 988 |
| | VCNeF | 8935.28 | $\pm 7.08$ | 26662 |
| | VCNeF sequential | 29063.89 | $\pm 79.58$ | 668 |

Table 18: Inference times and GPU memory consumptions of different models trained and evaluated on the 1D Burgers' equation with a spatial resolution of 256, predicting different numbers of timesteps in future.

## G  QUALITATIVE RESULTS

Here we provide a comparison of visualizations of the predictions vs ground truth for 1D Advection, Burgers, and 2D Compressible Navier-Stokes PDEs. The 2D CNS dataset has four channels, namely density, velocity-x, velocity-y, and pressure, and we visualize the predictions of our VCNeF model with the ground truth data.

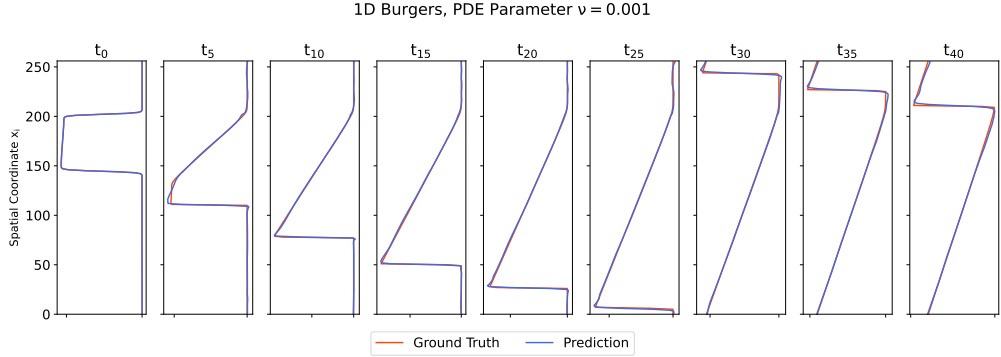

Figure 8: Example prediction's of VCNeF for 1D Burgers with $N_t = 41$ and $s = 256$ spatial resolution.

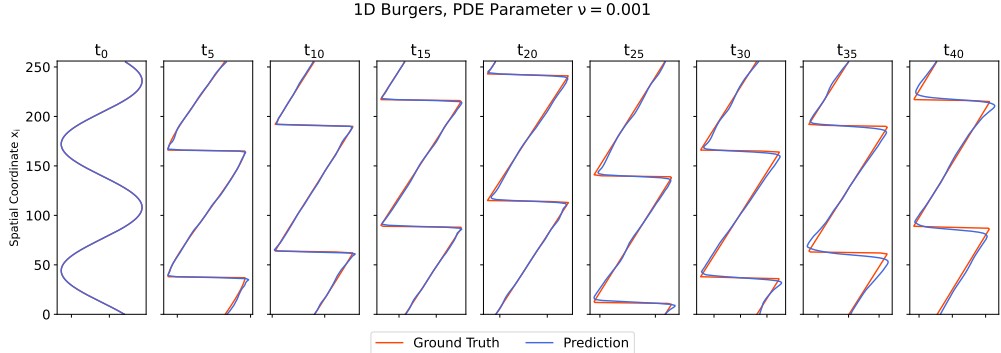

Figure 9: Example prediction of VCNeF for 1D Burgers with $N_t = 41$ and $s = 256$ spatial resolution.

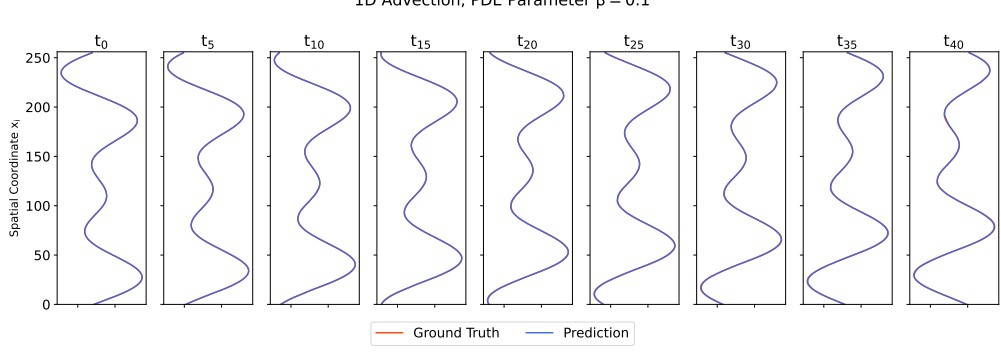

Figure 10: Example prediction of VCNeF for 1D Advection with $N_t = 41$ and $s = 256$ spatial resolution.

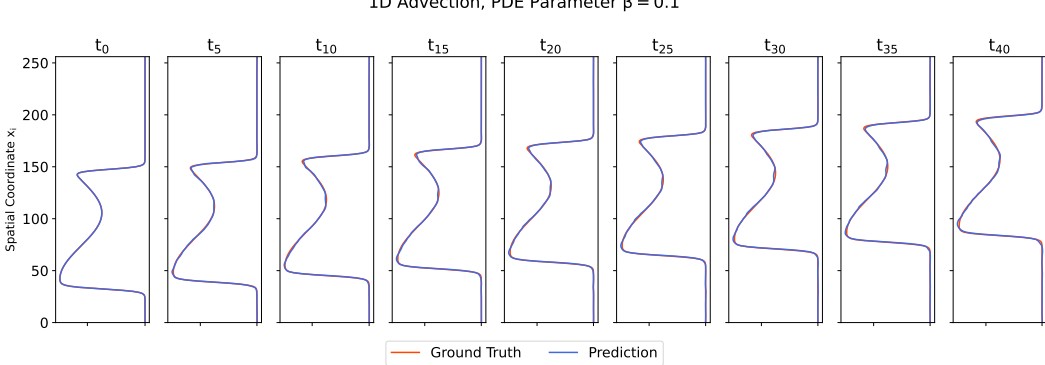

Figure 11: Example prediction of VCNeF for 1D Advection with $N_t = 41$ and $s = 256$ spatial resolution.

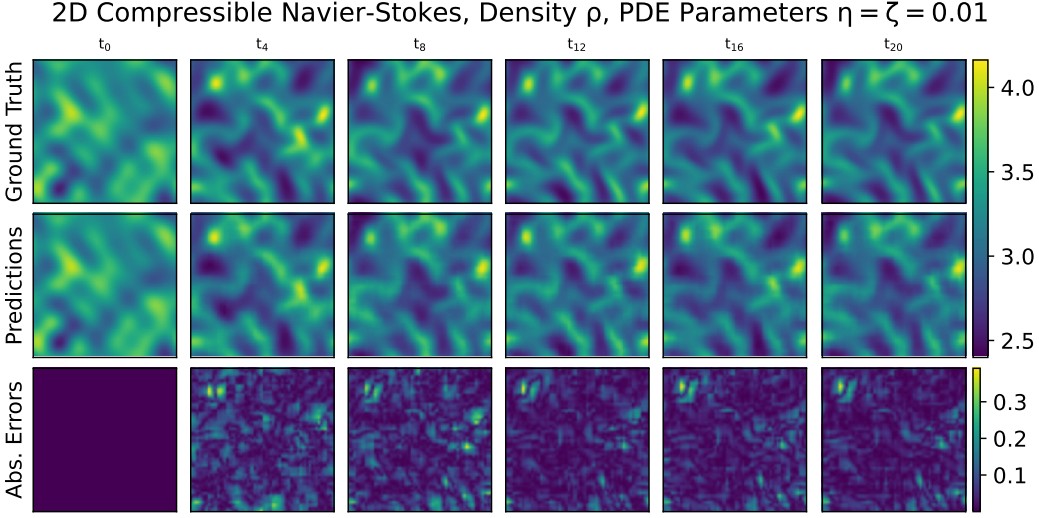

Figure 12: Example prediction of VCNeF for the density channel of 2D compressible Navier-Stokes with $N_t = 21$ and $64 \times 64$ spatial resolution.

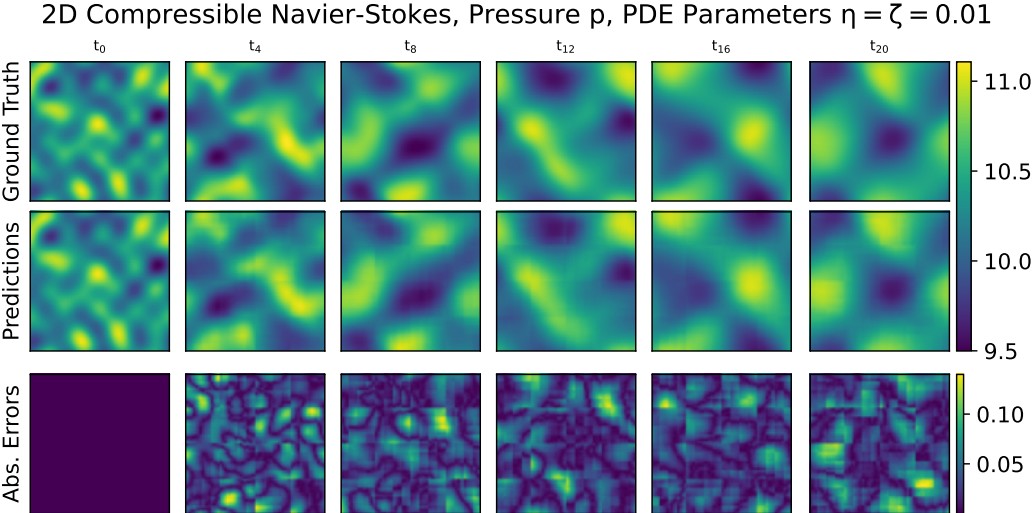

Figure 13: Example prediction of VCNeF for the pressure channel of 2D compressible Navier-Stokes with $N_t = 21$ and $64 \times 64$ spatial resolution.

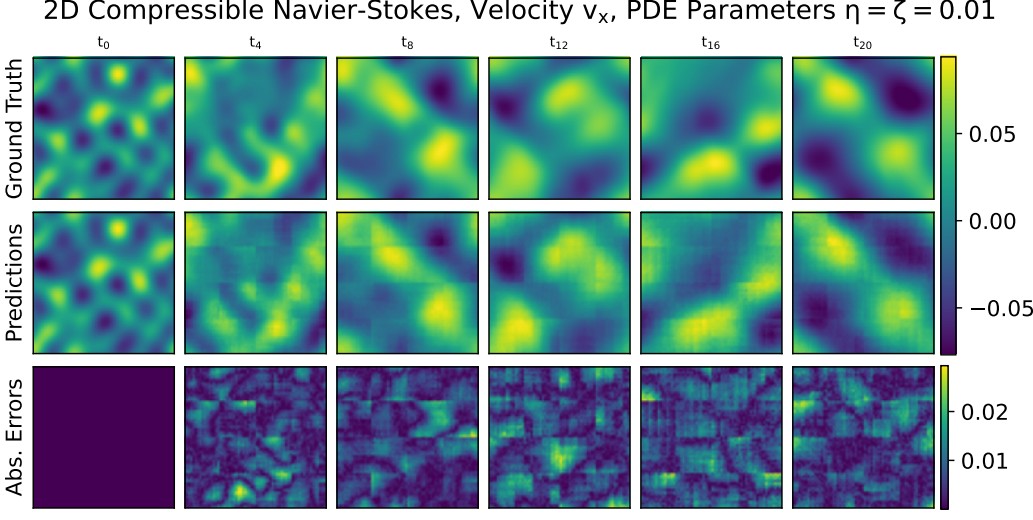

Figure 14: Example prediction of VCNeF for the velocity in x-direction of 2D compressible Navier-Stokes with $N_t = 21$ and $64 \times 64$ spatial resolution.

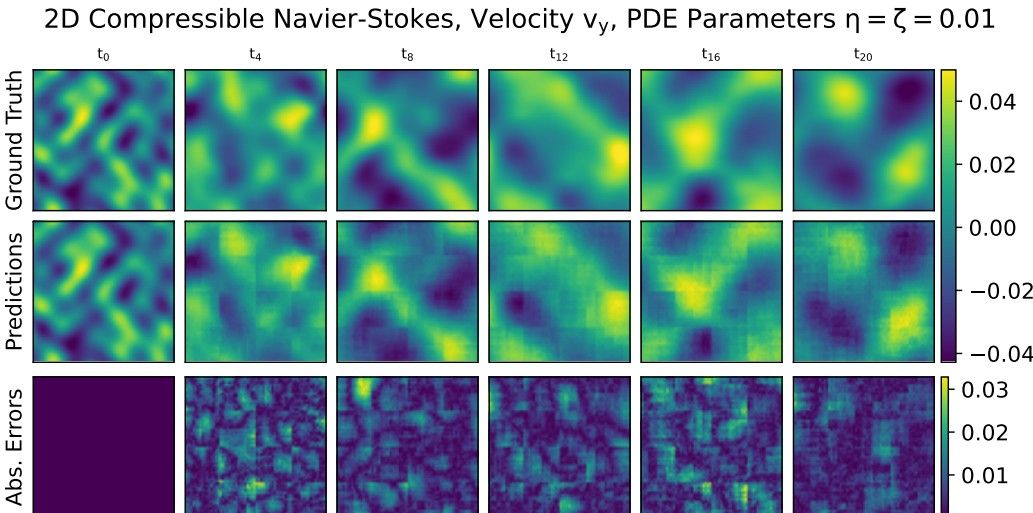

Figure 15: Example prediction of VCNeF for the velocity in y-direction of 2D compressible Navier-Stokes with $N_t = 21$ and $64 \times 64$ spatial resolution.

## H   TABLE OF NOTATIONS AND MATHEMATICAL SYMBOLS

### LIST OF MATHEMATICAL SYMBOLS

The below tabulation describes several symbols that are used within the article.

| | |
|---|---|
| $\mathbb{R}_+$ | Set of positive real numbers. Specifically, $t \in (0, \text{T}]$ |
| $\mathbb{R}^{s \times D}$ | Spatial grid of size $s$ and dimensionality $D$ |
| $\nabla$ | Gradient or vector derivative operator ($\frac{\partial}{\partial x}$, $\frac{\partial}{\partial y}$,) |
| $\Delta$ | Laplacian ($\nabla^2$) |
| $\partial_t$ | Partial derivative with respect to $t$ |
| $\partial_x$ | Partial derivative with respect to $x$ |
| $u(t, x)$ | Solution of the PDE at time $t$ for a given $x$ |
| $\mathbf{v}$ | Velocity vector field ($\mathbf{x}, \mathbf{y}, \dots$) |
| $\mathbf{p}$ | PDE parameter values, either a scalar or vector |
| $f_\theta$ | Neural network with learnable parameters $\theta$ |
| $N_t$ | Total number of timesteps in the simulation |

