# OpenReview forum: "Vectorized Conditional Neural Fields: A Framework for Solving Time-dependent PDEs"
_ICLR.cc/2024/Workshop/AI4DiffEqtnsInSci — AI4DiffEqtnsInSci @ ICLR 2024 Poster_

### Official Review · Reviewer_tiYc · 2024-02-27
**Potentially interesting idea with little to back it up experimentally**

**Rating:** 4
**Confidence:** 3

**Review:**

Verdict:
An interesting proposal with very little to back up the effectiveness of the method. The main novelties of this work seems to be: 1) the treatment of inputs and outputs as "neural fields" or in other words, inputs and outputs parameterized as functions. 2) vecrtorization of this neural fields 3) usage of transformers with these NN-valued inputs/outputs While 1 and 3 are not novel in itself see e.g. [1,2], I have not seen 2) yet. Where the paper breaks apart in my view is the sufficient evaluation and scientific documentation. The examples on which the architecture is evaluated are mere 1d toy examples. Moreover, parameter counts and hyperparameters are not disclosed, which makes it difficult to evaluate the effectiveness of the method is in comparison to the baseline models. Moreover, it is unclear to me what the trade-offs of this method are from the discussion presented in the paper.

For the above reasons, I am inclined to reject the paper.

Minor comments:
- I find the terminology "Neural fields" somewhat unnecessary in the realm of SciML. These are simply PDE solutions represented by Neural Networks. It's unfortunate that ML in general tends to favour trendy names such as Neural Fields, and would recommend sticking to clear-cut terminology wherever possible
- I would name it Appendix rather than tailpiece

References
[1] Yifan Du, Tamer A. Zaki - Evolutional Deep Neural Network (https://arxiv.org/abs/2103.09959)

[2] Louis Serrano, Lise Le Boudec, Armand Kassaï Koupaï, Thomas X Wang, Yuan Yin, Jean-Noël Vittaut, Patrick Gallinari - Operator Learning with Neural Fields: Tackling PDEs on General Geometries (https://arxiv.org/abs/2306.07266)

---

### Official Review · Reviewer_NHSb · 2024-03-01
**decent paper with a meaningful solution, but misses the literature and fair comparison with state of the art methods**

**Rating:** 7
**Confidence:** 3

**Review:**

This submission deals with fixing the caveats of transformer networks for solving PDEs. Some of the caveats include generalization to PDE parameters not seen during training,(ii) spatial and temporal zero-shot super-resolution,(iii) continuous temporal extrapolation, (iv) dimensionality generalization of PDEs, and (v) efficient inference for longer temporal rollouts. To solve these issues this work utilizes the attention mechanism to design conditional vectorized neural fields (VCNeF) that can achieve fast training and inference by parallelization on GPUs. The performance is compared with FNO and OFormer and it shows some gains.

This submission addresses an important and timely problem. The proposed solution is also meaningful. The comparison with FNO doesn't seem fair. There are new versions of FNO such as AFNO [1] based on transformer networks and attention that is scalable and fast to train. Also, SFNO [2] seems to improve the role out. I would recommend the authors to compare these two works and compare in the experiments  if possible.

[1] Guibas J, Mardani M, Li Z, Tao A, Anandkumar A, Catanzaro B. Efficient token mixing for transformers via adaptive fourier neural operators. InInternational Conference on Learning Representations 2021 Oct 6.

[2] Bonev, Boris, et al. "Spherical Fourier Neural Operators: Learning Stable Dynamics on the Sphere." arXiv preprint arXiv:2306.03838 (2023).

---

### Meta-Review · Area_Chair_dix6 · 2024-02-29

**Recommendation:** Accept (Poster)

**Metareview:**

The reviewers mention that this is an interesting approach, where the vectorization of neural fields is novel. I agree with Reviewer tiYc that the experimental results are limited and it is hard to draw conclusions on the effectiveness of the method but I feel that for preliminary work for the workshop it is acceptable. I also agree that the terminology Neural Fields is unnecessary. I vote for acceptance pending that the authors address the comments by the reviewers and add the references from Reviewer NHSb in the camera-ready version.

---

### Decision · Program_Chairs · 2024-03-01

Accept (Poster)